# Preparation of UV-Curable Nano-WO_3_Coating and Its Infrared Shielding Properties

**DOI:** 10.3390/nano12213920

**Published:** 2022-11-07

**Authors:** Zhengjie Wang, Rong Zhong, Ting Lai, Tianlei Chen

**Affiliations:** School of Environmental and Chemical Engineering, Nanchang Hangkong University, Nanchang 330063, China

**Keywords:** nano-tungsten oxide, IR shielding coating, UV-curable, high-energy ball-milling

## Abstract

Nano-WO_3_ particles are expected to find use in new shielding materials because of their significant absorption of near-infrared light in the 1400–1600 nm and 1900–2200 nm bands and high transmittance of visible light. In this study, WO_3_ was ground and dispersed using high-energy ball-milling to prepare a nano-WO_3_ dispersion using BYK331 as the dispersant and ethanol as the solvent. The prepared nano-WO_3_ dispersion was added to a photo-curing system and cured using UV irradiation to form films. The cured films were characterized using FT-IR, SEM, XRD, and TGA. The results showed that the nano-WO_3_ powder was evenly dispersed in the coating. The infrared blocking rate of the film continuously improved and the visible light transmission rate continuously decreased with increasing amounts of nano-WO_3_.For the film containing 6 wt%nano-WO_3_, the infrared blocking rate of the coating is 90%, the visible light transmittance is 70%, the hardness of the coating is 3B, and the adhesion is 3H. The thermal stability of the coating is also improved.

## 1. Introduction

Glass materials are used widely in the construction, electronic, automobile, and aeronautic industries [1]. Traditional window glass is usually composed of soda lime glass [2], which is made of silica, calcium oxide, sodium oxide, and a small amount of other additives through a float process [3,4,5]. The transmittance of transparent soda lime glass for near-infrared light at 4mm thickness is approximately 75–90% [6]. Near-infrared light (700–3000 nm) is the spectral region with the highest thermal energy [7], accounting for approximately52% of the total solar energy received on surfaces [8]. The high transmittance of glass for near-infrared light can easily cause temperature rise in internal spaces and produce light pollution [9,10,11,12]. Thus, materials for near-infrared light shielding have attracted the attention of many researchers and engineers.

At present, the most common infrared shielding materials are indium tin oxide (ITO) and antimony tin oxide (ATO), which can effectively block the near-infrared spectrum [13,14,15,16]. Khan et al. [17] prepared a multilayer nano-MgF_2_/ITO coating via electron beam evaporation. The composite coating has good infrared shielding ability, and its visible light transmittance can reach 90%. Wang et al. [18] prepared hollow ATO microspheres via a hydrothermal method with carbon spheres as templates. The ATO hollow microspheres have high specific heat capacity, high infrared emissivity, and low thermal conductivity, and effectively enhanced the photo thermal shielding performance of silicone acrylic lotion coating. However, the raw materials ITO and ATO can only shield wavelengths greater than 1500 nm [19,20], which limits their large-scale commercial applications. Some noble metal nanoparticles can also shield infrared light [21,22,23], but their utility in infrared shielding materials is limited by their low transmittance of visible light. Limited rare earth compounds, such as LaB_6_, PrB_6_, and NdB_6_ [24,25,26,27], can also effectively shield infrared light. However, they are expensive and large amounts of energy are required to crush their hard surfaces, making themun suitable for large-scale commercial production.The infrared shielding ability of cesium tungsten bronze is better than that of ITO and ATO, and it thus shows great application prospects in infrared shielding materials [28,29,30].Song et al. [31] prepared K-and C_S_-Co doped tungsten bronze (K_m_Cs_n_WO_3_, m + n = 0.32) via a solid-state reaction method. Their results showed that K-and C_S_-Co doping can promote the formation of W^5+^, enhance local surface plasm on resonance (LSPR), and improve near-infrared shielding performance at 730–1100 nm. The visible light transmittance of the materials reached66.89% and their near-infrared shielding rate reached 98.25%. Zhang et al. [32] prepared Li-C_S_-Co doped tungsten bronze nanoparticles with excellent near-infrared shielding properties via a simple solid-state reaction method. The results showed that Li^+^ doping can control the growth of Cs_x_WO_3_ crystals to form hexagonal nanorods with high aspect ratio and improve the near-infrared shielding performance. Tungsten-oxide-based materials have attracted attention because of their superior near-infrared shielding performance. However, there have been few reports of the direct dispersion of nano-WO_3_ into UV-cured coatings to achieve efficient infrared light shielding.

Nano-WO_3_ particles have small particle size, insufficient atomic coordination, large specific surface area, and high surface energy, which can cause the nanoparticles to exhibit high activity, instability, and easy agglomeration [33,34]. Appropriate physical or chemical methods [35,36,37] must be adopted to modify the nanoparticles to obtain nanoparticles with good dispersion, small particle size, and narrow particle size distribution. High-energy ball-milling is a new surface modification method that can significantly reduce reaction activation energy, achieve grain refinement, greatly improve the activity of powder, and improve particle size distribution uniformity, and enhanced with the combination of the interface between the body, and thus improve the compactness of the material, along with its electrical and thermal performance. This method is an energy-saving and efficient material preparation technology.

In this study, nano-WO_3_ is uniformly dispersed in a solvent using high-energy ball-milling. By changing the grinding and dispersion time, the dispersant, the proportions of the powder and dispersant, and the solvent, and examining the influence of these parameters on the dispersion system, a dispersive preparation process is developed. The obtained nano-WO_3_ dispersion is added to an UV-curable coating to prepare a nano-WO_3_-PUA composite coating. The aim of this study was to obtain an infrared shielding material for application to the surface of glass.

## 2. Experimental Section

### 2.1. Materials and Instruments

Nano WO_3_ (99.99%) was purchased from Aladdin Chemical Co., Ltd., Shanghai, China. BYK331 (50 wt%) was purchased from BYK additives Co., Ltd., Wesel, Germany.Ethyl acetate, ethanol, and n-butyl acetate were purchased from Guangzhou Xinguan Chemical Co., Ltd., Guangzhou, China. Polyurethane acrylate (B-619w) was purchased from Guangzhou BOXING Chemical Co., Ltd., Ghuangzhou, China. Monomer (HDDA) was purchased from Taiwan Changxing Chemical Co., Ltd., Guangzhou, China. The photoinitiator (184) was purchased from Tianjin JIURI New Materials Co., Ltd., Tianjin, China.

### 2.2. Preparation of Nano-WO_3_Dispersion

The desired amounts of dispersant and solvent were added to the mixer, and the stirring speed was set to 2500 rpm.After 30 min of stirring, nano-WO_3_ powder was added, and stirring was continued for an additional 30 min. The stirred dispersion was placed into the ball-milling apparatus and ground using 0.8 mm zirconium beads at a grinding speed of 3800 rpm for a grinding time of 4 h.Subsequently, 0.1 mm zirconium beads were used at a grinding speed of 3800 rpm for a grinding time of 4 h. Samples were taken every hour and a nanoparticle size meter was used for particle size measurement.

### 2.3. Preparation of Light Curing Coatings

The preparation of light curing coatings is shown in Figure 1. A formula containing the resin, monomer, and photoinitiator was used as the organic component of the UV-curable system. The proportion of B-619w was approximately 80% (*w*/*w*), the proportion of HDDA was 15%, the proportion of 184 was 5%. To this, 2wt%, 4wt%, 6wt%, and 8wt% nano-WO_3_ was added, respectively, and the mixture was diluted to 50% solid content with ethanol. The mixture was coated onto the glass with a 20# wire rod, baked in an oven at 80 °C for 1min, and then plated into a curing machine for curing.

### 2.4. Characterization

#### 2.4.1. Particle Size Analysis

The desired amount of the nanodispersion was placed in a beaker and diluted 20–30 times with toluene, stirred to uniformity. The diluted dispersion was taken up with a pipette and added to the cuvette, which was then placed into the particle size analyzer. The particle sizes were measured. Each sample was measured in three groups with ten measurements in each group.

#### 2.4.2. Infrared Spectrum Analysis

The cured film was peeled off from the substrate and ground into powder. Samples were prepared using the KBr tablet method and measured using a Nicolet Magna380 infrared spectrometer. The absorption peaks appearing at different positions were used to analyze the structure and composition of the substance.

#### 2.4.3. Field Emission Scanning Electron Microscope (SEM)

A small amount of the sample was evenly coated on the silicon wafer using a rubber dropper, dried, and solidified into a film. The sample was placed in a Nova SEM 450 field emission scanning electron microscope for scanning.

#### 2.4.4. X-ray Diffraction (XRD)

The coatings were characterized using a D8 advance X-ray powder diffractometer from Bruker, Germany. A portion of the sample was scraped from the cured coating and ground. A small amount of ground solid powder was placed in the instrument for scanning, The working voltage was set to 40 kV, the working current to 40 mA, the scanning angle to be 10–80°, and the scanning speed to be 3°/min.

#### 2.4.5. Thermogravimetric Analysis (TGA)

A small amount of film was scraped off the surface of the cured layer with a blade and then characterized using a Perkin-Elmer Diamond TG/DTA thermal analyzer in an argon atmosphere from 25 °C to 800 °C at a heating rate of 10 °C/min.

#### 2.4.6. Haze Analysis

The cured films were placed in a haze tester to test their haze. Haze measurements were taken at three positions, and the average value was taken as the haze of the layer.

#### 2.4.7. Absorbance Analysis

The cured coating film was placed horizontally in an LS160 transmittance meter. Upper, middle, and lower positions of the coating were measured, and the IR blocking rate and visible light transmission rate of the coating were recorded. Finally, the average value of the upper, middle, and lower positions was taken as the transmittance of light at different wavelengths for the layer.

## 3. Results and Discussion

### 3.1. Effect of Grinding Time on Particle Size of Nano-WO_3_

It can be seen from Figure 1a that when 0.8 mm zirconium beads are used for grinding, due to the large particle size of the grinding medium, the kinetic energy of the grinding medium is relatively large at a specific grinding speed, and the grinding effect of nanoparticles is relatively good. Therefore, the particle size of nano-WO_3_ decreases rapidly. However, when the grinding time exceeds 4h, the effective collision between the WO_3_ particles and the grinding medium is reduced due to the large gap between the grinding media and the small particle size of the nano-WO_3_, so the particle size of the nano-tungsten oxide no longer decreases. With continued grinding, the internal temperature of the system continues to rise, the solvent is volatilized, the viscosity of the system increases, the nanoparticles cannot be effectively dispersed, and agglomeration occurs again, and the particle size tends to increase, which may cause the system to become gelatinous and cause grinding failure.

After grinding had been performed using 0.8 mm zirconium beads for 4 h, the 0.1 mm zirconium beads were replaced and grinding was continued(Figure 1b). Due to the reduction in the size of the grinding medium, the number of beads will increase for a given amount of medium. The gaps between the beads will be reduced, so the grinding and dispersion of nano-WO_3_ will be more thorough, and the particle size of nano-WO_3_ will continue to decrease. However, when grinding was performed for more than 4 h using the 0.1 mm zirconium beads, the particle size increased.

Therefore, during the experiment, the slurry was sampled every hour to test the slurry particle size, and when the decreasing trend in the size slows or it begins to increase, the initial grinding medium should be replaced with smaller beads to make the grinding more efficient.

### 3.2. Effects of the Amount of Dispersant

Figure 2 shows the change in the particle size for nano-WO_3_ to dispersant ratios of 1:1, 1:2, and 2:1. During the first two hours of the grinding process, the average nano-WO_3_ particle sizes of the three groups are similar. However, beyond two hours the reduction in the nano-WO_3_ particle size for the WO_3_:BYK331 = 2:1 group is slower than that of the WO_3_: BYK331 = 1:1 group, and the final average particle size for the 2:1 group is larger. This is because, as the grinding progresses, the size of the nanoparticles continues to decrease, and the specific surface area of the particles continues to increase. When the amount of dispersant is small, it is difficult for the organic groups of the dispersant to completely coat the nano-WO_3_ particles. The tendency of the particles to agglomerate due to intermolecular covalent bonds or the van der Waals force becomes larger, and their dispersion becomes more difficult. For the WO_3_:BYK331 = 1:2 group, due to the high-speed rotation of the grinding medium during the grinding process, a large amount of heat is generated, resulting in a sharp rise in the internal temperature of the grinding system. The failure of the condensation system to immediately cool down will lead to a large amount of volatilization of organic solvents in the system. Moreover, the effective components in the dispersant will gelatinize, resulting in an increase in the viscosity of the system. The whole slurry will become colloidal, making it difficult to disperse. As shown in the figure, when nano-WO_3_:BYK331=1:1, the system is relatively stable and the grinding effect is the best.

### 3.3. Effects of Solvent on the Dispersion of Nano-WO_3_

To explore the influence of different solvents on the dispersion of nano-WO_3_, experiments using ethyl ester, ethanol, and butyl ester were conducted. The samples were ground using 0.8 mm zirconium beads for 4 h, and there after with 0.1mm zirconium beads for an additional 4 h, after which their particle sizes were tested. As shown in Figure 3, the particle sizes in the dispersion systems using different solvents were similar throughout the different stages of the grinding process, and the final particle size difference after grinding was less than 10nm. Therefore, it can be concluded that the choice of solvent has little effect on the dispersion effect of nano-WO_3_.

### 3.4. Particle Size Distribution of Nano-WO_3_Dispersion

The results of the above experiments indicate that the optimal conditions for the WO_3_ dispersion process are ethanol as the solvent in a proportion of 50%, nano-WO_3_ is 25%, and the BYK331proportion of 25%. The sample was ground with 0.8mm zirconium beads for 4 h and thereafter ground with 0.1mm zirconium beads for a further 4 h at a grinding speed of 3800n/min. The particle size diagram of nano-WO_3_ after grinding and dispersion is shown in Figure 4. After grinding and dispersion of the nano-WO_3_, the D50 is 54 nm and the D90 is 73 nm, the particle size distribution is narrow and the particle size is uniform.

### 3.5. FT-IR Testing of Composite Coatings

The FT-IR spectra of coatings prepared using different amounts of nano-WO_3_ (0%, 2 wt%, 4wt%, 6 wt%, and 8 wt%) are shown in Figure 5. The stretching vibration of O-W-O is the peak at 810 cm^−1^. The peak at 750 cm^−1^ is the characteristic absorption peak of W-O. The peak at 3375 cm^−1^ may be the characteristic absorption of W-O-H. The spectra exhibit the characteristic vibration peak of C=O at 1725 cm^−1^ for polyurethane methyacrylate. The absorption peak at 2931 cm^−1^ may be the characteristic absorption peak of -CH_2_ in the system. With increasing WO_3_ content, the intensity of the absorption peaks increased. These proved that WO_3_was successfully introduced into the coatings.

### 3.6. Morphology Analysis of Composite Coatings

Figure 6a presents the morphology of the coating prepared using unmodified WO_3_ and Figure 6b presents the morphology of a coating prepared using milled WO_3_. The figure shows that when nano-WO_3_ is introduced into the UV-curing coating, serious agglomeration occurs, while the milled nano-WO_3_ is evenly dispersed in the coating without obvious agglomeration. The sizes of the nano-WO_3_ are approximately between 20–200 nm.

### 3.7. XRD Analysis of Composite Coating

The XRD patterns of the coatings prepared using different amounts of WO_3_ dispersions(0, 2%, 4%, 6%, and 8%) are shown in Figure 7. The spectra of the WO_3_-containing coatings show two sharp absorption peaks at 2θ of 23.43°and 24.38° that are not observed for the coating without WO_3_, and with increasingWO_3_addition the intensities of these diffraction peaks and their peak areas increase. A weak diffuse reflection is observed at 33.72°, and it becomes sharper with the addition of more WO_3_. These indicate that the WO_3_ phase is present in the layer, and that as the amount of added WO_3_increases, the crystallinity of the crystalline phase becomes higher.

### 3.8. Thermal Stability Analysis of Composite Coatings

Figure 8 presents the TG data for the coatings prepared using different amounts of WO_3_, and Figure 9 presents the DTG data. As shown in Figure 8, he decomposition process of the coating at 0~800 °C can be roughly divided into two stages: 60–98 °C and 98–344 °C. Figure 9 shows that the maximum weight loss peak of the coating occurs at approximately 400 °C. The weight loss in the first stage is mainly due to the volatilization of residual solvent and free water in the system, and the weight loss rate is approximately 2%. The weight loss in the second stage is due to the decomposition of organic matter in the photo-curing system such as resin and monomer. The 6% WO_3_coating shows the best thermal stability. Overall, the decomposition process of coatings prepared using different amounts of WO_3_ is roughly the same, and the addition of WO_3_ has little effect on the thermal stability of the coatings.

### 3.9. Haze and Mechanical Properties of Composite Coatings

As presented in Table 1, the addition of nanoparticles to the coatings markedly improves the hardness, but negatively affects their adhesion and, with increasing nano-WO_3_ content, the haze of the film gradually increases. For 6% WO_3_, the hardness of the coating can reach 2H, the adhesion of the coating can reach 3B, and the haze of the coating is 1.8. Thus, the coating has excellent scratch resistance, excellent adhesion, low haze, and good transparency. When the amount of dispersion is 8%, the hardness of the coating can reach 3H, but the adhesion of the coating is 0B.

### 3.10. Optical Properties of Cured Coating

In Figure 10, the left and right ordinates represent the visible light transmission rate of the coating and the infrared light blocking rate of the coating, respectively. When no nano-WO_3_ is added, the visible light transmittance of the coating is 96%, and the infrared blocking rate is 3%. When 2% nano-WO_3_ dispersion is added, the visible light transmittance of the coating is 88%, and the infrared blocking rate is 46%. With the addition of increasing amounts of nano-WO_3_ dispersion, the visible light transmittance of the coating film gradually decreases, and the infrared blocking rate gradually increases. When the addition amount is 6%, the visible light transmittance is 70%, the infrared blocking rate is 90%, and the thermal insulation performance of the coating film is excellent. When the addition amount is 8%, the infrared blocking rate is 96%, the visible light transmittance is 62%, and the clarity of the coating film decreases.

## 4. Conclusions

In this study, the high-energy ball-milling method was used to grind and disperse nano-WO_3_, and the nano-WO_3_was added to an UV—curing system to prepare UV-curable organic–inorganic composite films. The results showed that the organic–inorganic composite UV-curable coating doped with nano-WO_3_ had excellent infrared barrier performance, and excellent mechanical and optical properties. When 6 wt% nano-WO_3_ was added, the visible light transmittance of the coating is 70%, the infrared barrier rate is 90%, the hardness of the composite coating is 3H, and the adhesion is 3B. Using high-energy sphere method to modify inorganic particles has many advantages. It can not only significantly reduce the reaction activation energy, but also refine the grains, greatly improve the powder activity, and improve the particle distribution uniformity. Compared with other modification methods, this method is simple and efficient. This product could find potential applications in glass for windows, automobiles, and aircraft.

## Data Availability

The data presented in this study are available on request from the corresponding author.

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
