# Peer review of "Preparation of UV-Curable Nano-WO3Coating and Its Infrared Shielding Properties"

_nanomaterials, 2022, doi:10.3390/nano12213920_

Round 1

Reviewer 1 Report

The submitted manuscript deals with modern topics and could therefore be published. However, I have three serious reservations that must be answered.

1. Interpretation and evaluation of FT-IR measurements. The authors write that a new W-O band appears at about 3300 cm-1 in the FT-IR spectrum. This is a wrong interpretation, because with heavy atoms the vibrations are manifested in the fingerprint region. Rather, I would advise the authors to consider whether water molecules bind to the material.

2. Authors must provide information about the substrate on which their material was deposited, as well as the thickness of the layer and the overall layer with the substrate.

3. Authors must provide information from which statistical file their data is. Whether it was just some initial experiment, or whether their results are reproducible. They should then support the statistics with the data obtained.

Author Response

Dear Reviewer:

On behalf of my co-authors, we appreciate you very much for your positive and constructive comments and suggestions on our manuscript entitled“Preparation of UV curable Nano WO3 coating and its infrared shielding properties”

The following are the responses. Thank you very much.

  1. Interpretation and evaluation of FT-IR measurements. The authors write that a new W-O band appears at about 3300 cm-1 in the FT-IR spectrum. This is a wrong interpretation, because with heavy atoms the vibrations are manifested in the fingerprint region. Rather, I would advise the authors to consider whether water molecules bind to the material.

Response: Interpretation and evaluation of FT-IR measurements were rewrited as followed. “The FT-IR spectra of coatings prepared using different amounts of nano-WO3(0%, 2 wt%, 4wt%, 6 wt%, and 8 wt%) are shown in Figure 5. Stretching vibration of O-W-O bonds are ascribed to the peaks at 810 cm-1 is, responsible for the distinctive peaks of WO3. The addition of WO3 to the coating give rise to new characteristic absorption peaks at 3375-1 and 750cm-1 also(M. Farhadian, P. Sangpour, G. Hosseinzadeh, A.K. et al., RSC Adv. 6 (2016) 39063), which are the characteristic absorption peaks of W-O bonds. The spectra exhibit the characteristic vibration peak of C=O at 1725cm-1 for polyurane methyacrylate. The absorption peak at 2931cm-1 may be the characteristic absorption peak of - CH2 in the system.

  1. Authors must provide information about the substrate on which their material was deposited, as well as the thickness of the layer and the overall layer with the substrate.

Response: The substrate is glass. The mixture was coated onto the surface of glass. The thickness of cured film is about 5 micrometer.

  1. Authors must provide information from which statistical file their data is. Whether it was just some initial experiment, or whether their results are reproducible. They should then support the statistics with the data obtained.

Response: These data were obtained through a lot of experiments and these results are reproducible. Our production process and formula have been provided to a factory to produce UV curable coating.

Reviewer 2 Report

The article 'Preparation of UV curable Nano WO3 coating and its infrared shielding properties' is well written from scientific background. I rise few comments before it goes to next level.

1. did the authors find any changes in chemical composition with grinding hours? 

2. how did the authors confirm chemical compositions of WO3? 

3. Since this is a oxide materials so oxygen vacancy might play a significant role in nanoparticle design, characterizations, and application test. What they think on this part?

Author Response

Dear Reviewer:

On behalf of my co-authors, we appreciate you very much for your positive and constructive comments and suggestions on our manuscript entitled“Preparation of UV curable Nano-WO3 coating and its infrared shielding properties”

The following are the responses. Thank you very much.

Comment 1: Did the authors find any changes in chemical composition with grinding hours?

Response:We did not find the changes in chemical composition with grinding hours.

Comment 2:How did the authors confirm chemical compositions of WO3?

Response: After grinding with different sizes zirconium beads, different sizes nano-WO3 can be prepared. In theory, the chemical compositions of WO3 was WO3-x, the x is between 0-1. In our experiments, By FT-IR, XRD, SEM and naked eyes observation, the chemical compositions is WO3 at room temperature. The morphology of the nanoparticle was close to spherical and blue. While the shape of WO3 was rod, the WO3 color is purple.

Comment 3:Since this is a oxide materials so oxygen vacancy might play a significant role in nanoparticle design, characterizations, and application test. What they think on this part?

Response: The existence of oxygen vacancies in materials can not only absorb near-infrared light, form a bridge for electron transmission at the position below the conduction band, but also absorb more oxygen molecules and convert them into active species. In the study, we design the nanoparticle to change the particle size for improve the light shielding function. And we try to use the nanoparticle to the UV curing system to prepare the films for the application of light shielding.

Reviewer 3 Report

comments are in the attached file: review of nanoWO3.docx

Author Response

Please see the attachent.

Round 2

Reviewer 1 Report

The authors partially improved the manuscript, but three fundamental things still remain. The first is a totally wrong interpretation of the FTIR data. In the region around 3350 cm-1 there can be no vibration of the W-O bond. Atoms are too heavy for that. The authors provided a citation to follow. However, if we look at the given reference, everything is correct there, but the nonsense that a band would arise at 3350 cm-1 is not mentioned in it. Therefore, the authors themselves should read this reference more carefully and try to explain the emerging band at 3350 differently.

The second thing is that the authors say they have a lot of results. So why don't they provide statistics (standard deviations). I mean especially in Figure 10.

And thirdly. The question of the total thickness of the sample is still not answered, i.e. glass plus WO3 film.

Author Response

Dear Reviewer:

Thank you very much for your helpful comments. Our responses as followed:

1: The first is a totally wrong interpretation of the FTIR data. In the region around 3350 cm-1 there can be no vibration of the W-O bond. Atoms are too heavy for that. The authors provided a citation to follow. However, if we look at the given reference, everything is correct there, but the nonsense that a band would arise at 3350 cm-1 is not mentioned in it. Therefore, the authors themselves should read this reference more carefully and try to explain the emerging band at 3350 differently.

Response :That reviewer` comment of the peaks around 3350cm-1 can be no vibration of the W-O may be right. It is very possible that the surface of WO3 may be produce a new bond of W-O-H or absorb a little H2O in the system. So the interpretation of peak around 3350cm-1 for WO3 was deleted.

2: The second thing is that the authors say they have a lot of results. So why don't they provide statistics (standard deviations).

Response: The results of the optical experiments showed that the standard deviations were small. For example, when the content of WO3 is 2%, the standard deviations of the visible light transmittance of this sample was 1.

.3: And thirdly, the question of the total thickness of the sample is still not answered, i.e. glass plus WO3 film.

Response: The glass thickness is 1.1mm, and the thickness of coating is 5μm. And the total thickness of the sample is 1.105mm.

Reviewer 3 Report

The authors have improved the manuscript a lot. Still, the English language needs attention.

Please correct the construction of the following sentences because they are not clear:

3.5 "Stretching vibration of O-W-O bonds are ascribed to the peaks at 810 cm-1 is, responsible for the distinctive peaks of WO3."

" The addition of WO3 to the coating give rise to new characteristic absorption peaks at 3375-1 and 750cm-1 also.which are the characteristic absorption peaks of W-O bonds"

"With increasing of WO3 addition, the corresponding absorption intensitive becomes larger, which proved that WO3 is successfully introduced into the coatings."

Fig. 4 - sign the figure as in the description in the text on page 7

Fig. 5 - correct English in the figure caption

Fig. 6 - in figure caption give description about part a) and b)

Figs 8, 9 - in the caption the word "coating" is missing

After these corrections the manuscript can be published.

Author Response

Dear Reviewer:

Thank you very much for your comments. Our responses as followed:

1:Please correct the construction of the sentences because they are not clear;

3.5 "Stretching vibration of O-W-O bonds are ascribed to the peaks at 810 cm-1 is, responsible for the distinctive peaks of WO3."

"The addition of WO3 to the coating give rise to new characteristic absorption peaks at 3375-1 and 750cm-1 also. which are the characteristic absorption peaks of W-O bonds"

"With increasing of WO3 addition, the corresponding absorption intensitive becomes larger, which proved that WO3 is successfully introduced into the coatings."

Response: "Stretching vibration of O-W-O bonds are ascribed to the peaks at 810 cm-1 is, responsible for the distinctive peaks of WO3" was corrected “ The stretching vibration of O-W-O is the peaks at 810cm-1”.

" The addition of WO3 to the coating give rise to new characteristic absorption peaks at 3375-1 and 750cm-1 also. which are the characteristic absorption peaks of W-O bonds" was corrected “ The peak at 750-1cm is the characteristic absorption peaks of W-O.”

"With increasing of WO3 addition, the corresponding absorption intensitive becomes larger, which proved that WO3 is successfully introduced into the coatings." Was corrected “With increasing of WO3 content, the intensity of the absorption peak increased.”

2: Fig 4. sign the figure as in the description in the text on page 7

Response: The figure has signed as in the description in the text on page 7.

3:Fig.5-correct English in the figure caption;

Response:The English title of Figure 5 is changed to “Infrared spectra of coatings with different WO3 additions”.

4:Fig.6-in figure caption give description about part a) and b);

Response:” (a) the coating using unmodified WO3, (b) coating using milled WO3” was added to the figure 6 caption.

5:Figs8, 9-in the caption the word "coating" is missing.

Response: In figure 8 and 9, the “coating” was added.

Round 3

Reviewer 1 Report

The reason why I wanted to know the whole thickness of sample was clear.

Authors introduced LS160 Transmission meter for the measurement of absorption and transmission of IR radiation, but this transmission meter has limitation for thickness 0.8 mm. And authors had the whole thickness 1.1 mm.

This discrepancy must be explained.

Then it is possible to publish this manuscript.

Author Response

Dear Reviewer:

Thank you very much for your comments. Our responses as followed:

1. Authors introduced LS160 Transmission meter for the measurement of absorption and transmission of IR radiation, but this transmission meter has limitation for thickness 0.8 mm. And authors had the whole thickness 1.1 mm.

Response: The number of glass thickness provided was wrong. The glass thickness is 0.1mm, not 1.1mm. The total thickness of the sample is 0.105mm.